

# Tsunami Hazard assessment and Scenarios Database for the Tsunami Warning System for the coast of Oman

Íñigo Aniel-Quiroga[1], José A. Álvarez-Gómez[2], Mauricio González[1], Jara Martínez Sánchez[1], Laura M. Parro[2], Ignacio Aguirre-Ayerbe[1], Felipe Fernández[1], Raúl Medina[1], Sultan Al-Yahyai[3].

[1]Environmental Hydraulics Institute "IH Cantabria", Universidad de Cantabria, Santander, Spain.
[2] Complutense University of Madrid, Department of Geodynamics, Stratigraphy and Paleontology, Faculty of Geology,/
[3] Directorate General of Meteorology and Air Navigation, DGMAN. Public Authority for Civil Aviation, PACA, Muscat, Oman

*Correspondence to*: Íñigo Aniel-Quiroga (anieli@unican.es)

**Abstract.** Advances in the understanding of tsunami impacts allow developing products to assess its consequences in tsunami–
prone areas, as it is the case of the coast of the Sultanate of Oman. This paper presents the followed methodology and the obtained results for the assessment of the tsunami hazard of the coast of Oman and the development of the scenario database that feeds its Tsunami Warning System (TWS). Initially, a seismo-tectonic analysis of the area was carried out, focused on identifying the seismic areas whose earthquakes could generate tsunamis affecting the coast of Oman. A database of 3181 tsunamigenic sources was characterized by means of the parameters that define their focal mechanisms. This database includes
scenarios with magnitudes *Mw* ranging from 6.5 to 9.25 within the study area, but it is especially focused on the Makran Subduction Zone (MSZ). The 3181 cases were numerically propagated to feed the database and to work as precomputed scenarios for the TWS: In case of tsunami, the results for the closest precomputed scenario (in location and magnitude) are shown. From the database, 7 worst-case scenarios were selected and computationally simulated at national and local scale, in 9 municipalities all along the coast of Oman, resulting in tsunami hazard maps containing relevant variables in the flooded
area, such as the inundation water depth and the drag level (hazard degree for people instability).
Finally, in order to manage conveniently the results, an online tool, called Multi-Hazard Risk Assessment System (MHRAS), was developed. This tool is a viewer that contains an easy-to-use application, including the results of the tsunami hazard assessment and the tsunami scenario database, and the selection algorithm to choose the proper case among the precomputed ones. The results of this research are part of the National Multi-Hazard Early Warning System of Oman (NMHEWS).

## 1.  Introduction

Tsunamis are relatively infrequent phenomena but they represent a greater threat than earthquakes, hurricanes and tornadoes. They have caused the loss of thousands of human lives and extensive damage to coastal infrastructures around the world. Recent tsunami events like those occurred in Japan in 2011 and in Chile in 2010 and especially, the tsunami of 2004 in the



Indian Ocean, have fostered the elaboration of mitigation strategies on those countries that are located in tsunami-prone areas. This is the case of the Sultanate of Oman.

The Sultanate of Oman is placed on the southeast corner of the Arabian plate. Its area covers 310.000 km$^2$ and its population is ca. 4.000.000 inhabitants (NCSI, 2014). The coast of Oman has approximately a length of 2.000 km and a great part of the

population lives in the coastal area. Muscat, the capital of Oman, is placed in the north coast of the country. From a seismic point of view, Oman is mostly affected by the Makran trench, on the opposite side of the Gulf of Oman (see Fig. 1). Large earthquakes along the Makran Subduction Zone (MSZ) have generated destructive tsunamis in the past (Heidarzadeh et al., 2008). Although the historical record is incomplete, it is believed that tsunamis from this region have significantly impacted on several countries bordering the Northern Arabian Sea and the Indian Ocean. A recent example is the tsunami generated

along the MSZ on November 28th, 1945. Its effects were noted in Pakistan, Iran, India and Oman. Run-ups between 5 and 10 meters were recorded (Heidarzadeh et al., 2009). This tsunami was caused by an $M_w$ 8.1 magnitude earthquake in the eastern segment of the MSZ, being an interplate thrust event that ruptured approximately 20% of the length of the subduction zone. Its epicenter, off of the Makran coast, was located at a focal depth of 25 km. Some authors defend that this earthquake generated a landslide triggering the tsunami, which then arrived delayed to the coast of Oman (Heidarzadeh and Satake, 2017).


Thus, occasional major earthquakes can be expected which have the potential to generate tsunamis that could be particularly destructive in the Gulf of Oman, and as a consequence, the tsunami events database that feeds the tsunami warning system and the tsunami hazard assessment of the coast of Oman were addressed in this study.

The objective of the **tsunami warning systems** is to generate products to warn people about the possibility of an arriving tsunami and to understand the emergency itself. These products include warning messages, graphics, procedures, exercises, and dissemination systems focused on ensuring their effectiveness during a tsunami warning.

These systems commonly contain a database of pre-computed tsunami scenarios. This database is built with numerical models and cover all the possible events affecting the area of interest. When a new earthquake occurs, the system presents the closer

pre-computed scenario, estimating relevant variables like the tsunami wave height and its travel time. The elaboration of the database for the tsunami warning system of Oman required the analyses of all the seismic structures whose earthquakes could generate tsunami affecting the coast of Oman, and the accurate characterization of the focal mechanism of the obtained tsunamigenic sources. Among them, the worst cases were selected and used to assess the tsunami hazard at National and Local scale.


The main objective of **the tsunami hazard assessment** (THA) is the accurate estimation and characterization of the coastal region that would be flooded in case of tsunami. This objective is commonly tackled by calculating the variables or parameters that describe the flooding in the study area, such as tsunami wave height, water depth onshore, hazard degree for people





instability, etc (Álvarez-Gómez et al., 2013; Lorito et al., 2015). A key value to determine is the maximum run-up, i.e. the maximum elevation to which water from a tsunami wave will rise during its flooding process. The optimal methodology to calculate the flooded area in case of tsunami is the application of validated numerical models to simulate the phenomena and its associated processes: generation, propagation and inundation.

These numerical models are commonly applied following two alternative but complementary approaches (Álvarez-Gómez et al., 2013): Probabilistic Tsunami hazard Assessment (PTHA) and Deterministic Tsunami Hazard assessment (or Scenario Based, SBTHA). When PTHA is performed, it is often considered as an extension of probabilistic seismic hazard assessment (PSHA) (Annaka et al., 2007; Burbidge et al., 2008; González et al., 2009; Grezio et al., 2010; Power et al., 2013; Sørensen et al., 2012), obtaining seismic return periods for potential tsunamigenic earthquakes and incorporating the random uncertainties on the fault and tidal level parameters. Deterministic analyses (SBTHA) are based on worst-case scenarios, where the maximum potential tsunamigenic earthquakes are simulated. In addition, an aggregated analysis combining the results obtained for a number of worst-case scenarios of tsunamis in an area can be done. In this study, a deterministic approximation has been applied. This approximation provides essential information for coastal planning, engineering and management in terms of security concerns (Tinti and Armigliato, 2003).

The THA results are represented in maps that contain the calculated variables, characterizing the flooded area. These results were obtained at both national and local scale. National scale maps allow identifying the most exposed areas in case of tsunami, as they contain information about the maximum Run-up and the inundation length onshore. On the other hand, local scale maps, allow tackling higher resolution studies, evacuation plans and mitigation strategies. The local assessment covered 9 of the main municipalities and coastal areas of Oman: Sohar, Wudam, Sawadi, Muscat, Quriyat, Sur, Masirah, Al Duqm, and Salalah., and at least 20 km at each side of these municipalities. The 9 local study areas comprise, altogether, 17 wilayat (administrative divisions in Oman) and 315 towns.

Once the tsunami hazard is assessed, the study of the potentially flooded area is commonly followed by an assessment of its vulnerability and risk, which can be approached from several points of view: human, economic, infrastructures, environmental, etc. This paper tackles and evaluates the tsunami hazard in Oman. The methodology to assess the tsunami risk, its link to disaster risk reduction, and its application to the case of Oman are addressed in a companion paper (Aguirre Ayerbe et al., 2018).

The paper is structured as follows: Section 2 describes the characterization of tsunamigenic sources affecting the coast of Oman. In section 3, the applied numerical model is presented together with the used topobathymetric grid domains and the output variables that were calculated. In section 4, the results of the numerical simulations, and their representation as hazard



maps are detailed for both the scenarios database and the deterministic THA, and the viewer developed for the representation of the results is explained. Finally, in section 5, a discussion and some conclusions of the study are drawn.

## 2. Seismic Sources


In order to calculate the database of tsunami scenarios and to assess the tsunami hazard for the coast of Oman, first a seismotectonic study was carried out. This study allowed evaluating the potential seismic structures whose earthquake could generate tsunamis affecting the coast of Oman. In this section, the main aspects of the tsunamigenic sources' characterization are explained.


### 2.1 Characterization of Tsunamigenic Sources for the Scenarios Database

In order to generate the seismic scenarios database an analysis of the main aspects of the seismic process related to the tsunami generation was tackled: First, the tectonics of the area, to characterize the major active faults and its potential rupture dimensions; and second, the focal mechanisms parameters, to characterize the geometric properties of the occurred
earthquakes.

The characterization of the seismicity is supported by two main aspects. Firstly, the historical and recent distributions of earthquakes, and, secondly, the focal mechanisms reflecting the rupture characteristics of the earthquakes in the last decades. These two aspects allow establishing "a priori" areas of higher hazard, as well as the macroseismic characteristics of the tsunamigenic earthquakes. The focal mechanisms allow to statistically constrain the spatial orientation of the earthquake
ruptures (strike, dip and rake), for the different tectonic zones defined.

Based on the results of both analyses a sources database for the different tectonic zones of the area was developed. From the tectonic characterization, the maximum potential rupture dimensions and the rheological characteristics of the lithosphere involved in the seismic deformation were obtained. Some of the rupture parameters and their variability can be constrained from the analysis of the focal mechanisms and the past earthquakes.
Based on these analyses 11 seismic areas were defined (Fig. 1). This seismotectonic study allowed defining and characterizing the tsunamigenic sources affecting the coast of Oman.

In order to obtain the potential earthquake magnitude that a fault can generate it is common the use of empirical scaling relations. These equations relate the magnitude of the potential earthquake with known dimensions of the fault. Such relationships have been studied for earthquakes in continental crust (e.g. Stirling et al., 2002; Wells and Coppersmith, 1994)
and are the basis for many seismotectonic studies and seismic hazard analyses. The variability of the relations is high, not only





between different rheological behavior crusts but also between different regions and tectonic settings (Dowrick and Rhoades, 2004; Strasser et al., 2010a)

Depending on the rheological characteristics of the tectonic zones, and on the type of earthquake rupture, different scaling relations were used (Blaser et al., 2010; Leonard et al., 2014; Strasser et al., 2010a), and occasionally different relations were

used for the same structures in order to check the sensitivity of the results.

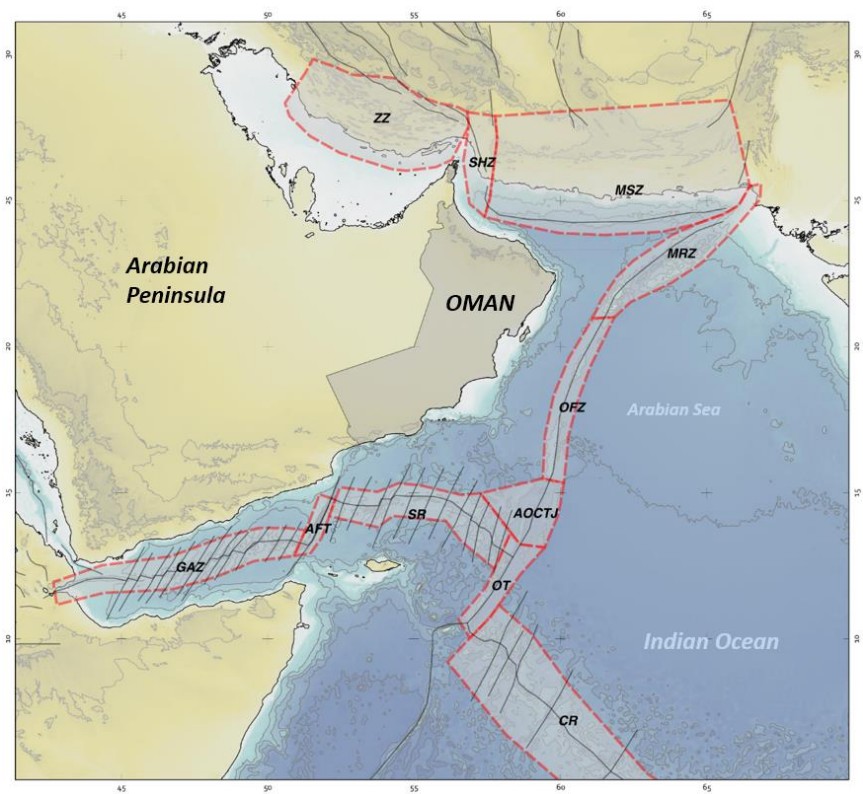

**Fig. 1. Tectonic zonation of the study area. ZZ: Zagros Zone; SHZ: Strait of Hormuz Zone; MSZ: Makran Subduction Zone; MRZ: Murray Ridge Zone; OFZ: Owen Fracture Zone; AOCTJ: Aden – Owen – Carlsberg Triple Junction; GAZ: Gulf of Aden**
**Zone; AFT: Alula – Fartak Transform; SR: Sheba Ridge; OT: Owen Transform; CR: Carlsberg Ridge.**

Once the earthquake magnitudes were related to the dimensions of the faults, and knowing the geometrical characteristics of the fault planes (from the tectonics and the focal mechanisms analysis) a series of tsunamigenic seismic sources for events

with magnitudes greater than 6.5 were defined.

Most of the potential tsunamigenic sources are located in the Makran subduction, where the greatest earthquakes in the area can be generated. A complete set of scenarios with magnitudes ranging from 6.5 to 9.25 were defined. These scenarios are



spatially distributed covering all the potential earthquake locations in order to contemplate the maximum number of possibilities for the early warning system.

This approximation is different from the use of "unit" sources   (e.g. Gailler et al., 2013). Instead of scaling some parameters from predefined units, all the characteristics of the rupture for each scenario were define. The ruptures that were studied and characterized are (Fig.  1):

**Makran Subduction Zone (MSZ)**.

In the Makran subduction, two differentiated segments with apparently different seismic behaviors have been described. An eastern segment that has presented relevant seismic activity in historical times, with large subduction events, like the earthquake in 1945 (Heidarzadeh et al. 2009); and a western segment without significant seismicity in recent times. However, (Shah-Hosseini et al., 2011 obtain results in their analysis of coastal boulders that are only explainable if the western Makran segment behaves in a seismic manner and generates large tsunamigenic events, although less frequently than the eastern

segment. Similarly, Mokhtari et al. (2008) describe Holocene raised marine terraces (<10 000 years) along the Makran coast, both east and west, that are typical markers of recent tectonic activity. In a simple way, the absence of large earthquakes in recent times can be explained by two hypotheses. Or the subduction interface does not have coupling (i.e., does not accumulate elastic energy and therefore cannot generate earthquakes), or is coupled and it is storing this elastic energy to release it as a large earthquake in the future. As there is no conclusive data to rule out coupling and potential seismic behavior of the western

segment, and underestimating the maximum earthquake in the area, or exclude the occurrence of a major earthquake in the western segment, could cause thousands of casualties in the medium term and a serious lack of emergency planning, it was decided to include the western segment as a potential source in the same terms as the eastern segment. In MSZ, the worst-case scenario is then defined by the total length of the potential rupture of the whole subduction interface.

For the elaboration of the tsunami scenario database, the minimum magnitude of tsunamigenic earthquakes was considered Mw=6.5, implying a rupture length of 20 km. Accordingly, a potential earthquake epicenter each 20 km along the length direction was defined. Analogously the width predicted for a $M_w$=6.5 earthquake is approximately 24 km. As the total width of the subduction interface projection is 220 km a potential earthquake epicenter each 22 km was defined along the subduction interface width. From the magnitude $M_w$=6.5 to the worst-case $M_w$=9.25 magnitude a source for each potential epicenter was

defined while the rupture dimensions allow the source to be contained in the subduction interface. E.g. for the magnitude $M_w$=6.5, 404 rupture models were defined (Fig.  2a), while for the magnitude $M_w$=9.0, 38 potential ruptures were defined with slightly different strikes, epicenters and centroid depth (see Fig.  2b).





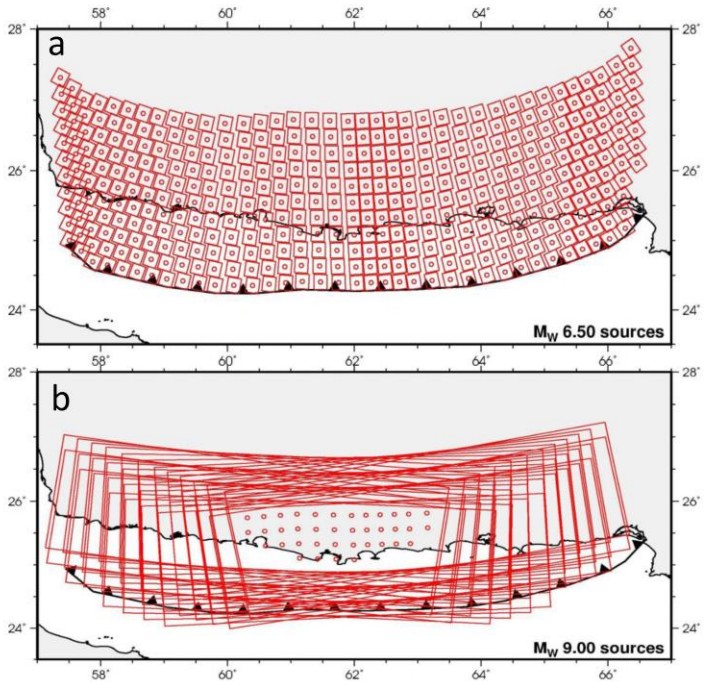

**Fig. 2. Epicenters (circles) and simplification of Mw=6.5 (a) and Mw=9.0 (b) magnitude earthquakes at Makran subduction zone.**

**Zagros Zone (ZZ).** The active Zagros fold-thrust belt lies on the northeastern margin of the Arabian plate, on Precambrian (Pan African) basement. This is a young (Pliocene) fold-thrust belt currently undergoing 10±4 mm yr⁻¹ shortening and thickening as a result of collision of the Arabian and central Iranian plates (Berberian, 1995; Vernant et al., 2004). The tectonic

mapping of Berberian (1995) and Blanc et al. (2003) were used to estimate the maximum earthquake based on the maximum mapped fault length. A maximum length of 110 km was obtained. According to the scaling relations of Blaser et al. (2010) or Wells and Coppersmith (1994), it is equivalent to an M 7.75 earthquake. In the Zagros Zone, only the frontal offshore thrust was modeled, but allowing magnitudes from 6.5 to 7.75. A total number of 148 ruptures along the Zagros front were modeled (see Fig. 3).



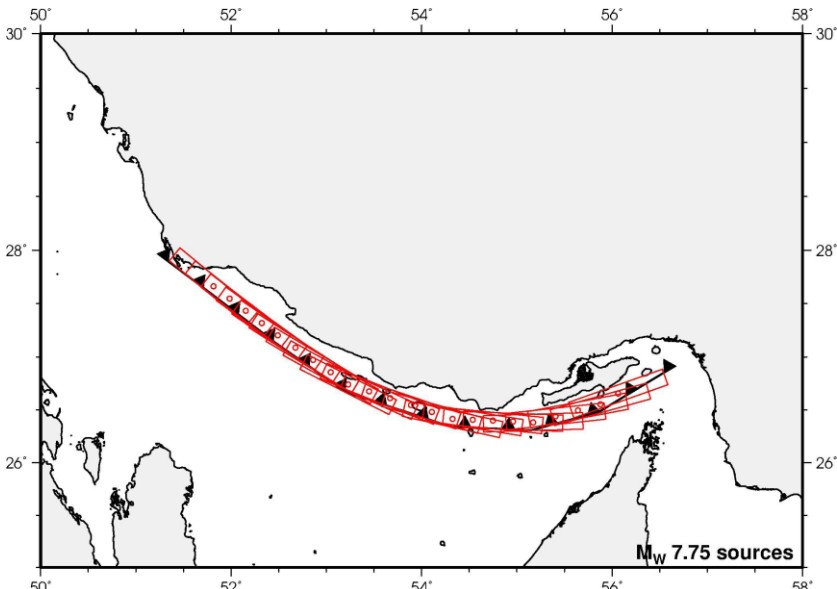

**Fig. 3. Epicentres (circles) and simplification of Mw=7.75 magnitude earthquakes at Zagros.**

**Strait of Hormuz Zone (SHZ)**. In the SHZ the NW trending Zendan-Minab-Palami fault system connects the Zagros fold-and-thrust belt to the Makran prism long a NNW trending fold-and-thrust belt, 20 km wide and 250 km long. It is oblique to the regional convergence and is characterized by strike-slip, oblique reverse and thrust faulting as well as fault propagation folds  (Regard et al., 2004). According to the characterization of the faults in the area, a conservative maximum rupture length of approximately 45 km was estimated. The strike is NNW-SSE with values from N150ºE to N180ºE and dipping towards the east with high angles. The fault motion is of oblique character, with right-lateral and reverse dip-slip components, but predominantly right-lateral. In addition to the strike-slip and oblique ruptures, some reverse faulting earthquakes are present in the area (Fig.  4). In order to take into account the possibility of this kind of earthquakes in the strait, some scenarios with the characteristics of these earthquakes were modeled. As mapped faults could not be used, as in the strike-slip structures described for the zone before, it was decided to constrain the maximum rupture length with the width of the strait. This maximum rupture length is 80 km in the N60ºE direction which can produce earthquakes with magnitudes Mw 7.5 according to the Blaser et al., 2010  relations, with a rupture width of 39 km. With this width and a dip of 25º (from the focal mechanisms) a maximum rupture depth of 16 km was obtained, which is coherent with the seismotectonics of the area. A total of 48 sources were modeled in SHZ including the oblique-strike slip and reverse events.





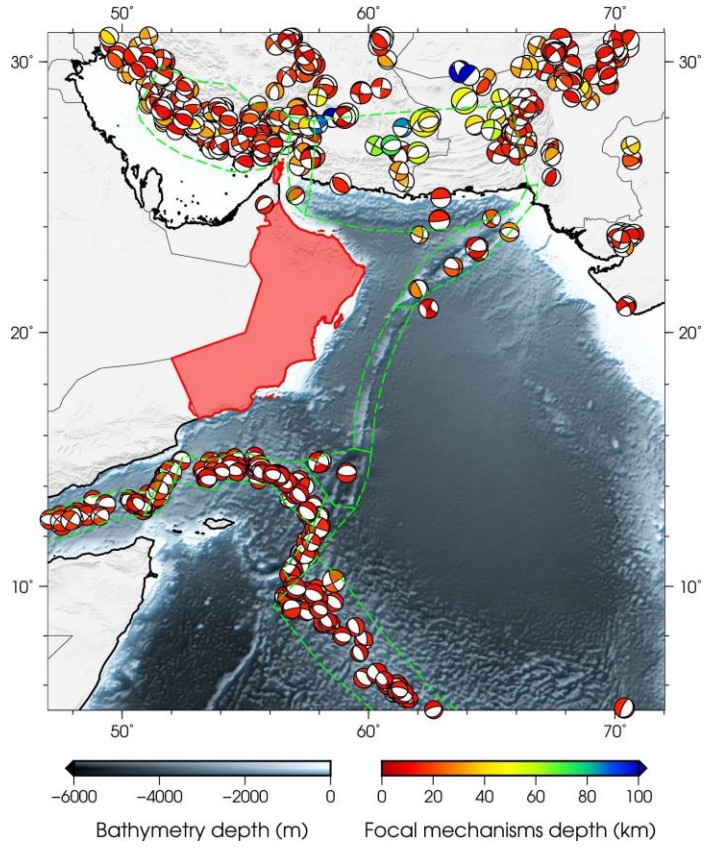

**Fig. 4. Focal mechanisms from the Global CMT catalog** (Ekström et al., 2012)**. The shading is function of the focal mechanism depth.**
**The defined seismotectonic zones are marked by dashed lines.**

**Murray Ridge Zone (MRZ)**. The Murray Ridge System is divided into basins and ridges. This has been interpreted as a transform margin of the Indian Plate that has been active since the Upper Cretaceous. Seismic reflection profiles from the Murray Ridge in the Gulf of Oman, show a significant component of extension across the predominantly strike-slip Indian–Arabian plate boundary. The Murray Ridge lies along the northern section of the plate boundary, where its trend becomes more easterly and thus allows a component of extension (Mokhtari et al., 2008). Due to the lack of detailed structural maps for the Murray Ridge a maximum earthquake similar to the defined by Deif et al., 2013 was used, modeling 33 ruptures of magnitude 6.5 following two families of fault planes, one striking N100ºE, dipping 60º towards the north with a rake of -135º (an oblique right-lateral dip-slip rupture); and other striking N35ºE and dipping 60º towards the NW with a rake of -135º (Fig. 4).

**Owen Fracture Zone (OFZ)**. The OFZ is a pure strike-slip boundary between the Arabian and Indian plates and is marked by a moderate seismicity and by a prominent bathymetric ridge, the Owen Ridge, up to 2000-m high with respect to the



surrounding seafloor. The Arabian plate moves northwards slightly faster than the Indian plate at a differential rate of 2 to 4
mm yr⁻¹ (Fournier et al., 2008). Commonly a strike-slip motion (horizontal displacement) is unable to generate dangerous
tsunamis; for this reason, only worst-cases scenarios were modeled, which in the area corresponds with the maximum length
of 180 km described by Fournier et al., 2011. This worst-case scenario is represented by a Mw ~ 7.75 earthquake according to
the Blaser et al. (2010) relations.

**Aden-Owen-Carlsberg Triple Junction (AOCTJ)**. At its both tips, the Owen Fracture Zone terminates into extensional
structures associated with basins. To the south, the OFZ terminates abruptly into the Beautemps–Beaupré Basin, a 50-km-wide
and 120-km-long basin bounded by two N70-N90°E-trending conjugate master normal faults (Fournier et al., 2001, 2008,
2011). The N70ºE normal faults dip towards the south and the N90ºE faults, on the southern edge of the basin, dip towards the
north. The maximum length of this normal faults was used in order to estimate the worst-case scenarios related to this structure.
Using the structural map of (Fournier et al., 2008) a maximum length for the northern faults of 46 km was obtained, and, in
the same way, 34 km for the southern faults was calculated, which corresponds approximately to earthquakes with magnitudes
6.75 and 6.5 (Blaser et al., 2010; Wells and Coppersmith, 1994). In the AOCTJ, the northern and southern normal fault
bounding the basin, and the strike-slip 37°30' E transform were modeled.

**Gulf of Aden Zone**. Situated between southern Arabia and the Horn of Africa, the Gulf of Aden links the Ethiopian rift and
the Red Sea with the Carlsberg Ridge in the NW Indian Ocean. The ridge crest is offset by numerous NNE-SSW-trending
structures identified as left-stepping transform faults with right-lateral motion. The Gulf of Aden is characterized by an oblique
opening. The present-day spreading direction is close to N25°E along the Alula-Fartak transform fault, as indicated by slip
vectors of earthquake focal mechanisms (Fournier et al., 2010). The obliquity is accommodated by an échelon faulting within
the axial rift, with normal faults oblique to the ridge trend (Fournier et al., 2010). The normal faults have directions of N100-
120ºE while the transform fault strikes N25-30ºE. The transform faults are short in general (< 50 km) and consequently, being
of pure strike-slip character, unable to generate relevant tsunamis. It was decided to discard these transform faults in the
western part of the Gulf of Aden as tsunamigenic. On the other hand, the normal fault segments present lengths up to 70 km,
but with the rupture confined to the first 5–10 km of depth in the lithosphere due to the thermal condition near the spreading
centers (Huang and Solomon, 1987). In this worst-case of normal faulting earthquake, assuming a reasonable dip of the fault
of 60º from the focal mechanisms, a maximum width of 11.5 km was obtained. This width, multiplied by the maximum length
of 70 km, gives a maximum rupture area of 805 km². Using different relations of Area vs. Mw (Leonard, 2010; Strasser et al.,
2010a; Wells and Coppersmith, 1994) a magnitude Mw 6.9 earthquake is obtained. Taking into account these results it was
decided to model 44 normal faulting earthquakes in the area from magnitudes Mw 6.5 to a maximum magnitude of 7.0 using
the self-consistent relations of  Leonard et al., 2014.





**Alula – Fartak Transform**. The Gulf of Aden Ridge, in the central part, is offset over 200 km by one major transform fault, the Alula-Fartak transform. This structure presents a maximum length of circa 200 km, which could generate a complete rupture, a magnitude Mw 7.75 similarly to the pure strike-slip faulting of the OFZ. This worst-case scenario was modeled in a similar way to the worst-case scenario of the OFZ.

**Sheba Ridge**. In the eastern part of the Gulf of Aden, the Sheba Ridge axis is offset by minor transform faults. The spreading rate along the eastern Sheba Ridge is currently slightly faster (2.4 cm yr$^{-1}$) than along the western Carlsberg Ridge (~2.2 cm yr$^{-1}$). Arabia is thus moving northward more rapidly than India with respect to Somalia. The Arabia-India relative motion was taken up by the Owen fracture zone as has been described previously. The Sheba ridge present similar characteristics to the Gulf of Aden ridge, and both parts of the ridge are usually studied as a whole. Being the tectonic characteristics similar, the same dimension constrains used for the Gulf of Aden was adopted. 65 rupture models with magnitudes from 6.5 to 7.0 were modeled for the Sheba Ridge.

**Owen Transform**. The Owen transform fault offsets by 330 km the Carlsberg Ridge and connects to the Sheba Ridge, which continues westward in the Gulf of Aden. As a transform fault its motion is almost pure strike-slip and is analogous to the Alula-Fartak transform fault described before. In the same way, only the worst-case was modeled, since the tsunamigenic potential of the strike-slip faults is very low. In the Owen Transform 3 scenarios of strike-slip type were modeled.

**Carlsberg Ridge**. The Carlsberg Ridge present similar characteristics to the other two main ridge axis in the area, the Gulf of Aden and Sheba ridge. The same dimension parameters and criteria described before were used. A total of 74 ruptures with magnitudes ranging from 6.5 to 7.0 of normal type were modeled in the Carlsberg Ridge.

In the following table the number of sources modeled for each zone is shown.

**Table 1. Number of sources considered at each seismic zone**

| Seismic Zone | Number of sources |
| --- | --- |
| Aden Gulf | 44 |
| Alula Fartak | 1 |
| AOCT | 14 |
| Sheba ridge | 65 |
| Owen transform | 3 |
| Carlsberg Ridge | 74 |



| Owen Fracture | 16 |
|---|---|
| Murray Ridge | 33 |
| Hormuz | 48 |
| Zagros | 148 |
| Makran | 2735 |
| **TOTAL** | **3181** |

The sources were described as simple rectangular ruptures. Their parameters were defined in a way that are easily incorporated into the numerical modeling of surface deformation (by means of the Okada model, Okada, 1985a). It was computed, for each one of the modeled sources: i) the hypocentral location (center of the earthquake rupture): longitude, latitude and depth; ii) the rupture orientation: strike, dip and slip rake; iii) the rupture dimensions: fault length and width; and iv) the average earthquake slip over the rupture plane.

**2.2 Tsunamigenic Sources for the Tsunami Hazard Assessment**

The tsunami hazard was assessed following a deterministic approach, i.e. by means of numerical simulations of worst-case scenarios. These scenarios were selected from the tsunami scenarios database created in the seismotectonic study presented in the previous subsection. The results of these scenarios were used to calculate and develop tsunami hazard maps, where several useful variables are represented for the study area. These maps are presented and explained in next sections.

From the 3181 cases of the database, 6 potential tsunamigenic sources were chosen and, additionally, the historical earthquake occurred in 1945 was included. The characteristics of the 7 selected sources that were used for the numerical simulation can be seen in Table 2.

**Table 2. Selected sources to elaborate the tsunami hazard assessment**

| Seismic Zone | Magn Mw | Lon | Lat | Depth (km) | Strike | Dip | Rake | Slip (m) | L (km) | W (km) |
|---|---|---|---|---|---|---|---|---|---|---|
| Zagros | 7.75 | 56.063 | 26.653 | 6.1 | 250.4 | 35 | 90 | 8.1 | 104 | 21 |
| Makran | 8.75 | 63.185 | 25.119 | 11.27 | 258.3 | 7 | 90 | 7.8 | 438 | 163 |
| Makran | 8.75 | 60.614 | 25.066 | 11.27 | 276 | 7 | 90 | 7.8 | 438 | 163 |
| Makran | 8.75 | 59.494 | 25.312 | 14.436 | 266.9 | 7 | 90 | 7.8 | 438 | 163 |
| Makran | 9.25 | 61.985 | 25.312 | 14.436 | 266.9 | 7 | 90 | 15.6 | 860 | 234 |
| Makran | 9.25 | 61.41 | 25.328 | 14.436 | 270.9 | 7 | 90 | 15.6 | 860 | 234 |
| 1945 | 8.1 | 63 | 24.5 | 20 | 240 | 5 | 90 | 7.5 | 126 | 38 |






As explained the Makran subduction zone has two segments, with an approximate length of 430 km each. By applying the empirical relations of Strasser et al., 2010b or Blaser et al., 2010, this length implies earthquake magnitudes of Mw 8.8. However, if both segments break in one earthquake, the rupture length would be about 850 km, implying maximum earthquake magnitude Mw ~ 9.25. The cases to be included in the tsunami hazard assessment were chosen from a conservative point of view, and, consequently, the possibility of the total break of both segments in one earthquake and the inclusion of Mw 9.25 cases was adopted.

The hazard mapping based on the potential occurrence of events of great magnitude, such as Mw 9.25, although conservative, and with return period of the order of thousands of years, allows defining a "flood peak" on land and the potential flood zones, which would be vital to define safety limits for evacuation zones, from a global risk assessment point of view. The experience of the 2004 tsunami in Indonesia and Tohoku, Japan, in 2011 have shown that unlikely events with return periods of thousands of years, can occur. Although the period of return is high, these events could take place at any time.

## 3.  Numerical simulation of the tsunami events

Numerical models are commonly applied to assess the tsunami hazard. These models use as input the tsunamigenic sources characteristics and the topobathymetry of the study area. The output of the numerical simulations are the variables that characterize the propagation and inundation processes of the calculated tsunami.

In this section, the numerical model used to conduct the tsunami simulations is explained. First, the model itself is exposed and afterward, the grid domains adopted for the calculations of scenarios database and for the THA, at both National and local scales, are presented. Finally, the variables that the numerical simulations give as output are defined and detailed.

### 3.1  Characteristics of the applied model

The numerical model used to simulate the tsunami scenarios was COMCOT ( Cornell Multi-grid Coupled Tsunami Model , Wang, 2009). COMCOT solves the nonlinear shallow water equations (NLSWE) using a leap-frog finite differences scheme on a 2D horizontal domain.

In a Cartesian coordinate system, the NLSWE can be expressed as follows:

*Mass conservation equation:*

$$\frac{\partial \eta}{\partial t} + \frac{\partial P}{\partial x} + \frac{\partial Q}{\partial y} = 0 \tag{1}$$

*Momentum conservation equations:*

$$\frac{\partial P}{\partial t} + \frac{\partial P^2}{\partial x} + \frac{\partial PQ}{\partial y} + gD\frac{\partial \eta}{\partial x} + \tau_x D - fQ = 0 \tag{2}$$

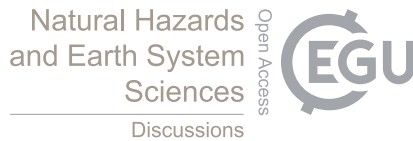

$$\frac{\partial P}{\partial t} + \frac{\partial PQ}{\partial x} + \frac{\partial Q^2}{\partial x} + gD\frac{\partial \eta}{\partial y} + \tau_y D - fP = 0 \qquad (3)$$

where η is the free-surface elevation above mean sea level; x and y represent the longitude and latitude of the earth; $\tau_x$ and $\tau_y$ are the bottom shear stress; P and Q stand for the volume fluxes ($P = Du$ and $Q = Dv$, with $u$ and $v$ being the depth-averaged velocities in the longitude and latitude direction); D is the total water depth (D = h + η), with h being the water depth; $f$ represents

the Coriolis parameter; and $g$ is the acceleration due to gravity. Numerical models based on LSWE and NLSWE are generally very efficient simulating tsunamis propagation.

The C3 model (Olabarrieta et al., 2011), a modification of COMCOT developed in the Environmental Hydraulics Institute of the University of Cantabria was adopted. Both, COMCOT and C3 are widely used and validated worldwide (Álvarez-Gómez

et al., 2013; Liu et al., 2009; Roshan et al., 2013; Wang and Liu, 2006, 2005).

Initial and boundary conditions are introduced defining a seafloor deformations from the seismic tsunamigenic sources defined in the previous subsection 2.2. They were modeled with the Okada equations (Okada, 1985b) included in the COMCOT-C3 model to transform the seafloor deformation into water surface deformation.


### 3.2 Grid Domains

Several data sources were used to build the bathymetric grids for the numerical simulations of propagation and flooding: (1) GEBCO bathymetry database (International Hydrographic Organization, 2014), (2) digitized bathymetric charts in near-shore areas and (3) high- resolution bathymetric data from hydrographic surveys. For the topography, a digital terrain model (DTM)

with a cell size of 10m provided by PACA (Public Authority of Civil Aviation of the Sultanate of Oman) was used.
These data were merged and 14 regular grids were elaborated, distributed in three nested levels. Several levels were nested to obtain an adequate resolution both offshore and near the coast. The characteristics of these grids (cell size, covered area) are given in Table 3. In Fig. 5, the area covered by each one of the constructed grids is depicted.

The first level, with lower resolution, was used not only to elaborate the tsunami scenario database but also to work as the biggest grid of the hazard assessment, where the rest of the grids were nested. This first level grid contained all the seismic areas affecting the coast of Oman. Level 2 grids were nested to level 1 grid and they were used to approach the propagations to the coastal areas by increasing the resolution of the bathymetry. The THA at National scale was conducted with the results of the numerical simulations obtained at this level 2 grids. Finally, one grid for each one of the 9 main coastal municipalities


of Oman was built at high resolution (Cell size around 45 m), and nested to level 2 grids. The THA at local scale was conducted
at this high-resolution local grids.

**Table 3. Schematization of nested grids.**

| Level | Grids | Area | Resolution | Application |
|---|---|---|---|---|
| 1 | 1 | Whole Coast | 1350 m | Hazard Assessment and scenario Database |
| 2 | 4 | Coastal areas | 270 m | Hazard assessment |
| 3 | 9 | Municipalities | 45 m | Hazard assessment |


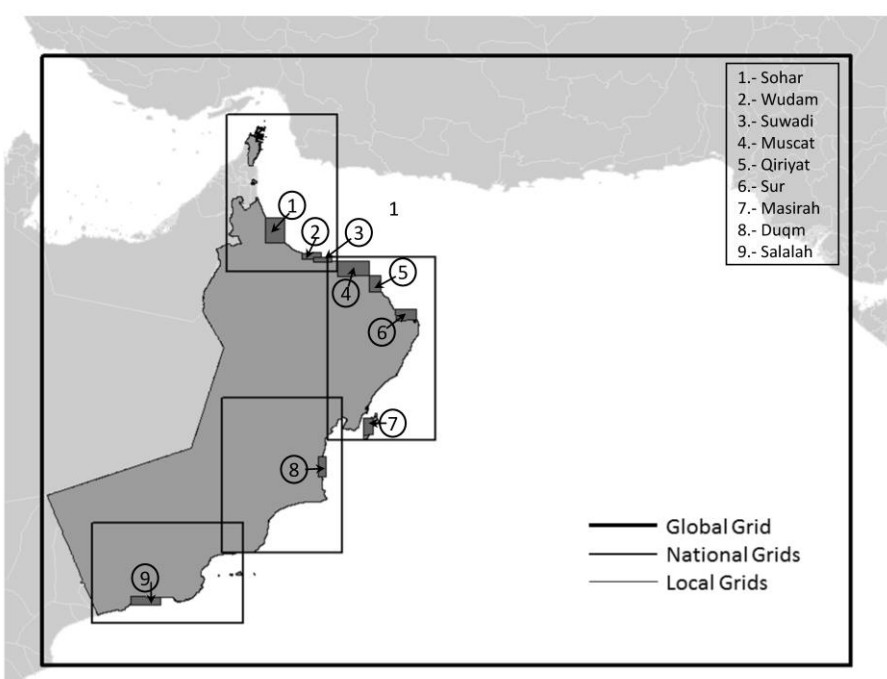

**Fig. 5. Distribution of grids: global grid (Level 1), National or approximation grids (Level2) and the 9 Local grids (level 3).**

### 3.3  Output Variables of the numerical model

As a result of the propagation, the model gives relevant variables that characterize the tsunami phenomenon, its propagations
and the inundation that triggers when it reaches the coast. The variables that are usually represented in tsunami hazard maps
are (see Fig. 6): Inundation depth (h), the vertical distance between the water surface and the ground (or the sea bottom);
Elevation ($\eta$), the amplitude of the wave from the sea level to the water surface; Run-up  (Ru), Maximum topographic
elevation reached by the tsunami in its flooding process; Inundation length (L), horizontal distance between the coastline and



the maximum flooding; Flow velocity (u) and Drag level (d), hazard degree for people instability, calculated as flow velocity

times the inundation depth (u·h).

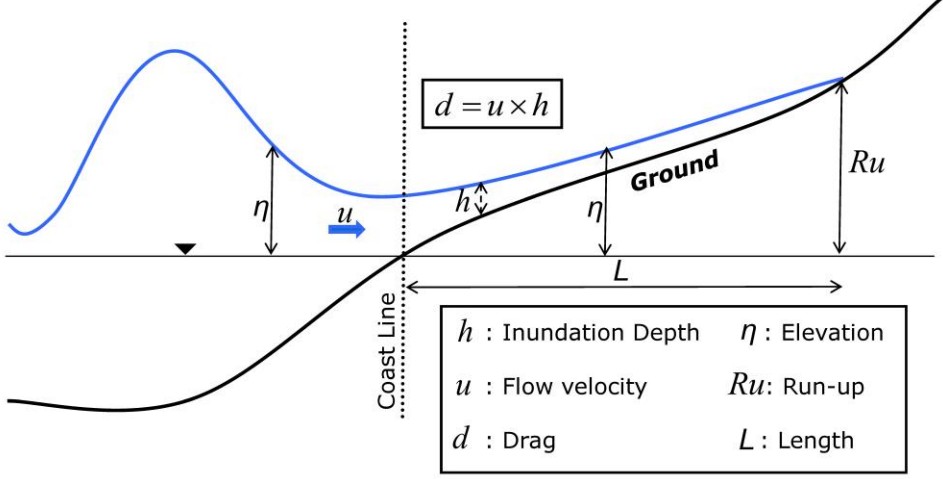

**Fig. 6. Scheme of the output variables obtained from the numerical simulations and represented in the THA maps**

## 4. Results

The objective of this research was twofold: on the one hand the characterization of the tsunami hazard over the entire coast of

Oman and focused on 9 municipalities, and, on the other hand, the computation of a tsunami scenario database.

### 4.1 Tsunami Scenarios Database and MHRAS

The numerical simulations of the 3181 cases that form the database were carried out by using the COMCOT model and for

each one of them, several variables were obtained. Thus, the generated tsunami database was used to calculate 2 variables: (1)

Tsunami maximum wave elevation data and (2) Minimum tsunami travel time of a detectable positive amplitude of 2 cm wave.

The results of all these numerical simulations are part of the tsunami warning system of the Sultanate of Oman. The calculated

variables were included as maps in the MHRAS (Multi-Hazard Risk Assessment System of the coast of Oman) (Aniel-Quiroga

et al., 2015; Fernández et al., 2014). For this MHRAS an easy-to-use online viewer was created (Fig. 7) to manage the

elaborated data. Using the epicenter locations (both numerical and graphical) and the earthquake magnitude as input, the

system automatically searches and selects the closer scenario among the 3181 precomputed cases, taking into account the

defined location and magnitude. Once the selection has been made, the tsunami travel time map and the tsunami maximum

wave elevation data are shown in the viewer (Fernández et al., 2014). In Fig. 7, an example of the maximum wave elevation

for an event with epicenter on Makran Subduction zone is shown.




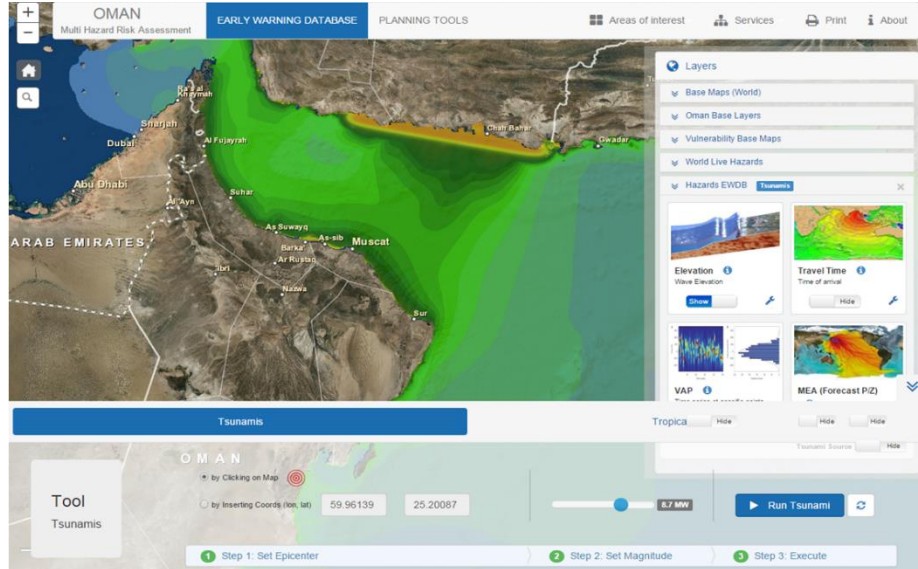

**Fig. 7. Extract of the MHRAS. Maximum wave elevation for a Mw=8.7 event, in Makran Subduction Zone**

In addition, for each one of the scenarios of the database, the tsunami maximum wave elevation time series was obtained at
specific locations: The validation points and the forecast points.

### 4.1.1    Validation points

The validation points are the actual locations where tidal gauges are placed on the coast of Oman. For each one of the tsunami
scenarios, the time series at these locations was obtained and depicted. In Fig. 8, an example of one of the graphs included in
the MHRAS is given. In the system, the graph is given by clicking on the validation point location once the scenario has been
chosen. These graphs allow knowing the time series on the coast in advance and also are useful to compare the computed and
real record data once the tsunami arrives the tidal gauge.





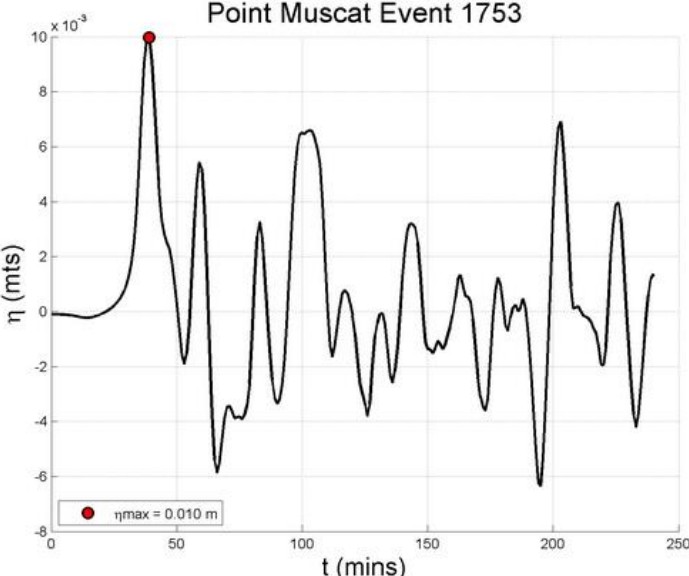

**Fig. 8. Example of the time series obtained for the location of the Muscat tidal gauge, including the maximum value of the tsunami wave elevation**

**4.1.2    Coastal Forecast Points (CFP) and Zones (CFZ)**

The Coastal Forecast Points (CFP) are points on the coast of each country at a depth of 30 m that work as key locations in a tsunami warning system. MHRAS is connected to the Indian Ocean tsunami warning system, and the location of 124 CFP on the coast of Oman and close countries was set by IOC-UNESCO in agreement with PACA of Oman (see Fig. 9a).





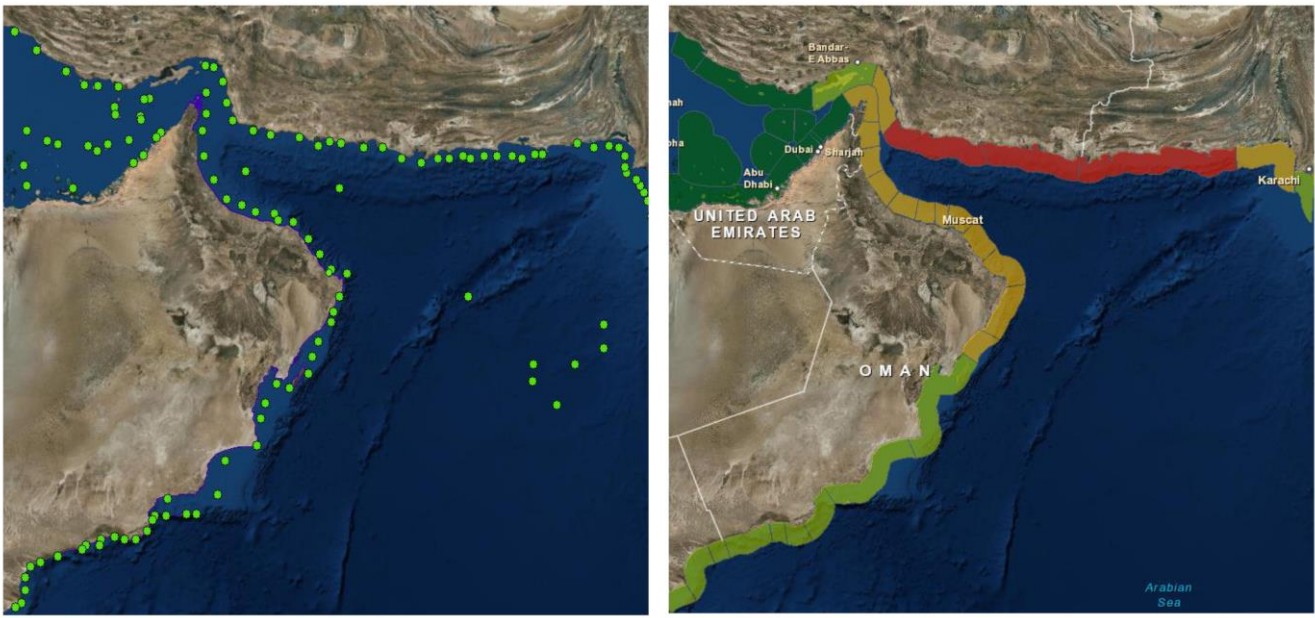


**Fig. 9. a Coastal forecast points defined by UNESCO where the time series of each tsunami scenario was obtained. b. CFP are grouped in CFZ and for each one the threat category was calculated for each tsunami scenario of the database and its representation was included in the MHRAS**

In order to categorize the tsunami hazard due to each scenario of the database at each CFP, a decision matrix was applied (see

Table 4), based on the values of the tsunami wave amplitude and the tsunami travel time from the source to the coastal area.

In this matrix 4 levels were defined: Tsunami Warning (red), Tsunami alert (orange), tsunami Watch (yellow) and threat passed

(green).

CFP were grouped in Coastal Forecast Zones (CFZ), defined as sections of 100 km of coastline. For each CFZ, the worst threat

category among its CFP was selected and depicted in maps like the one given in Fig. 9b. The CFZ maps allow knowing from

a unique map the hazard in the whole area.

**Table 4. Threat categories, calculated by means of the tsunami travel time from the source to the CFP and the tsunami wave amplitude.**

| Amplitude | T1<60 mins | T2>60 mins |
|---|---|---|
| >2m | Warning | Alert |
| 0.5-2m | Alert | Watch |
| 0.2-0.5 m | Watch | Watch |
| <0.2 | Threat passed | Threat passed |






These results, the obtained values at the validation points and forecast points and zones, were also included in the MHRAS viewer.

MHRAS is currently part of the Indian Ocean Tsunami Warning system, it receives relevant data regarding the tsunamigenic

sources, and selects the closer already-modeled case, sharing the maps that include the results of the propagations.

## 4.2 Tsunami Hazard Maps

The THA was carried out at National and local scale. At National scale run-up, inundation length and wave elevation were obtained from the numerical simulations. And at local scale drag level, inundation depth was obtained. The definition of this variables is given in Fig. 6.


The tsunami hazard maps were represented by means of aggregated maps. These aggregated maps (Álvarez-Gómez et al., 2013) combine the results of all the considered scenarios into one unique map. At each point of the map, the worst value of the represented variable among the calculated scenarios is given. I.e. if an elevation value of 4 meters is given at a point, this means that among the 7 studied scenarios the maximum elevation is precisely 4 m. This type of map allows representing

graphically the worst condition on every point in one unique map. Although these "worst" cases might have a low probability of occurrence, they are useful to represent the maximum flooding levels and to identify the most exposed areas, from a deterministic point of view.

### 4.2.1     Tsunami Hazard Map at National Scale

In Fig. 10, the tsunami hazard map of the coast of Oman at National scale is given. In this map, two variables were represented:

Run-up and Inundation length. The combination of these 2 variables gives an adequate view of the hazard, regardless the shape of the coastal area. I.e. in an area of cliffs the run-up can be very high, but complementing this information with a large or small inundation length, allows understanding better the tsunami scenario at each point of the coast. On the other hand, in a very flat area, the inundation length can be large, but knowing whether the run-up is high or low gives a more accurate view of the threat. In the National scale map both variables are given along the coast.

In this map it is observed that the highest tsunami hazard was obtained in the north coast of the country, as those tsunamis arriving from the Makran subduction zone reach Oman coast directly. On the east coast, there is a very flat area protected by Masirah Island, where the inundation length is large, but the obtained run-up is low. In the south of the country, the affection of the considered tsunamigenic sources is low, and the calculated run-up an inundation length produce a low tsunami hazard.



**Fig. 10. Tsunami Run-up (m) and inundation length (m) on the coast of Oman. Aggregated map. National scale.**

### 4.2.2    Tsunami Hazard Map at Local Scale

The 7 selected scenarios were modelled until the level 3 of the constructed nested grids (see Table 3), with a resolution of 45 meters at the 9 local municipalities (see Fig. 5). As a result, tsunami hazard maps containing two variables were developed focused on the characterization of the flooded areas.



The variables that were represented in the local maps are the inundation water depth and the drag level, using the equinoctial highest tidal level as reference.

In Fig. 11 the aggregated maps of maximum inundation water depth of the coastal area of the 9 municipalities in case of tsunami are given. In these maps it is observed that the hazard at those municipalities located in the north part of the country
(from Fig. 11 a to Fig. 11 f) is higher than the hazard at the eastern part, where the inundation water depth remained low, being almost negligible in Salalah (Fig. 11 i) and Duqm (Fig. 11 h). In the north, the maximum inundation depth in most coastal areas of Wudam (Fig. 11 c) and Suwadi (Fig. 11 d) would reach 5-10 m. And in Muscat (Fig. 11 a), the capital city of Oman, the inundation depth of the western part of the city, where the airport is located, would be also around 5-10 m. It is important to highlight that these are aggregated maps, which means that the worst situation among the 7 modeled scenarios at
each point of the municipalities has been represented.

In Fig. 12 the maximum drag level maps of the coastal area of the 9 municipalities in case of tsunami are given. In these maps, the represented variable combines the water depth and the flow velocity (Jonkman et al., 2008) to stablish a scale of hazard degree for people instability. In these maps, in accordance to the results of the inundation depth variable, the worst region for
people instability in case of tsunami is the north of the Country, while the east coast is barely affected.





**Fig. 11 a to 11 i. Maximum inundation water depth of local areas, in high tide situation. Aggregated maps**



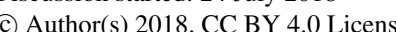



**Fig. 12 a to 12 i. Maximum drag level of local areas, in high tide situation. Aggregated maps**






In the manner of the results of the tsunami scenarios database, all the results of the THA were included in the MHRAS. This easy-to-use tool allows consulting any of the developed results. This viewer collects all the tsunami hazard maps, including all the calculated variables. The user just needs to select the scenarios (among the 7 modeled ones) and the scale (National or

Local) and the tool provides the final map. The aggregated maps, represented in Fig. 11 and Fig. 12, were also included in the HRAS viewer.

## 5.   Discussion and conclusions

The tsunami hazard assessment of the coast of Oman has been addressed by means of the calculation of a tsunami scenarios

database and the elaboration of tsunami hazard maps at National and local scale.

First, the potential seismic structures whose earthquake could generate tsunamis affecting the coast of Oman were studied and characterized. Although the main structure affecting the coast of Oman is the Makran Subduction Zone, other 10 seismic areas were analyzed in order to cover all the possible tsunami scenarios, including earthquakes with magnitudes from 6.5 to 9.25.

As a result, a number of 3181 scenarios were found and characterized and the parameters that define their focal mechanisms were set.

The 3181 tsunami scenarios were modelled, numerically, at national scale with the COMCOT-C3 model. These numerical simulations provided the maximum elevation maps and minimum tsunami travel time along the coastal area of Oman, and the

resulting propagations were included in an online viewer, the MHRAS (Multi Hazard Risk Assessment System). MHRAS is an easy-to-use tool that provides the results of the simulations in the form of maps. In order to visualize the results, the viewer also includes a selection algorithm. By providing the epicenter location and the magnitude of an earthquake, the system automatically makes available the maps of the closest scenario of the precomputed database. These maps show the maximum wave height on each point, and the minimum time required by the tsunami to reach. Obviously, tsunami travel time depends

on the proximity between the specific point that is being checked and the source of generation of the tsunami. On the other hand, the wave height mainly depends on the magnitude of the earthquake that has generated the tsunami. In general, the bigger the magnitude, the higher the wave. Besides, once the MHRAS has already selected the pre-computed case, the system allows the user consulting the numerically recorded data at several interesting points: forecast points and validation points. The forecast points are internationally agreed key points that, in case of a tsunami event would receive and then spread all the

relevant data, and validation points are the actual locations of tidal gauges.



Among the 3181 tsunami scenarios, the 7 worst possible events for the coast of Oman were selected to address the tsunami hazard. These 7 scenarios were modeled at national and local scale, including the calculation of the flooded coastal areas. At National scale, the run-up and the inundation length were calculated and represented in a National tsunami hazard map (see

Fig. 10). And, at local scale, the inundation depth and the hazard degree for people instability were calculated and mapped. Both National and local scale maps were elaborated from a deterministic point of view. The hazard maps at National and local scale of each one of the 7 chosen scenarios, as well as the aggregated case are included in the MHRAS viewer.

National Hazard map is focused on the identification of the areas that would be more affected in case of tsunami. In this map

(see Fig. 10), maximum run-up and horizontal inundation length can be observed. From the analysis of the National scale map, it can be said that the most affected area in case of tsunami would be the north coast of Oman, located in the gulf of Oman. This is due to its proximity to Makran subduction zone. Besides, there are some areas where the value of the maximum Run-up is quite high, but the inundation length is short, like in the case of Qiriyat area. This behavior is associated with cliffs areas. In the same way, some aisled areas with huge inundation length but low Run-up can be observed. This behavior is

related to very flat areas, like, for instance, the continental side in front of Masirah Island.

Local hazard maps of the 9 main coastal municipalities focused on the identification of specific affected areas in case of tsunami. The resolution of the grids was higher than in the case of National maps.

Analyzing these maps it can be observed that even for the aggregated maps, the envelope of all the worst cases, the eastern

coast of Oman would be very lowly affected. There are some areas where the horizontal inundation length is long, but the wave's elevation is very small. This occurs, for example, at the continental area of Masirah strait (see Fig. 11g), as it can be also observed in the national maps.

Again, the most affected areas are located in the north coast of the country, in the Gulf of Oman. The longest inundation was recorded at Wudam and Sawadi (see Fig. 11c and 11d), and in the same way, the highest elevation is observed in the west of

Muscat area (see Fig. 11 a). On the other hand, in Qiriyat (see Fig. 11f), due to the existence of high cliffs, the inundation area is smaller, and it can be said that just low zones or wadis (dry rivers) would be affected. The main affection comes from the Makran Subduction Zone events. The affection on the north coast by the Mw=7.75 event at Zagros subduction zone hardly affects the north coast, and the eastern coast is negligible. Just in the northern considered municipality (Sohar, Fig. 11 a), some affection could be expected from Zagros since it is the closest area to the Strait of Hormuz. In Duqm and Salalah (Fig.

11 h and Fig. 11i), on the eastern coast, the flooded area is very small, and again, just low areas and wadis would be flooded.

The results of the tsunami hazard assessment allow tackling the next steps in the tsunami risk assessment. Once the affected area is calculated, exposition and vulnerability assessments will provide the necessary framework to properly assess the tsunami hazard. Therefore, mitigation measures are proposed and applied. This is, from both theoretical and practical points

of view, the aim of a companion paper (Aguirre Ayerbe et al., 2018), focused on the disaster risk reduction.



**Acknowledgements**

The authors would like to thank the Ministry of Transport and Communications of the Government of the Sultanate of Oman (MOTC), Directorate General of Meteorology (DGMET), Public Authority for Civil Aviation (PACA), for supporting and
funding the project "*Assessment of Coastal Hazards, Vulnerability and Risk for the Coast of Oman*" and the collaboration of the IOC-UNESCO personnel.

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
