# Peer review of "Tsunami Hazard assessment and Scenarios Database for the Tsunami Warning System for the coast of Oman"

_Natural Hazards and Earth System Sciences, 2018_

## Referee Comment (RC1) · Anonymous Referee #1 · 21 Aug 2018

General Comments

This paper describes the development of a tsunami scenario database and its application for a tsunami warning system and hazard assessment for the coast of Oman. The work is an important contribution to the description of the tsunami hazard for Oman. The use of the same set of scenarios for the warning system and the hazard assessment has merit. While I can not comment on the seismological sections of the paper, that aspect appears to have been carefully assessed through the analysis of previously published work.

There are some gaps in the background information, some of the choices made need

to be justified, and there is a lack of verification. I have indicated below in 'Specific Comments' where I believe further effort may be required for the manuscript to be acceptable. The English is generally understandable, but the text needs some work. I have made a few suggestions for improvement below in 'Technical Comments' but am unable to do this for the entire manuscript. The final draft should be edited by a native speaker.

Specific Comments

1. I would recommend a slight change to the title. The publication essentially describes the development of a scenario database, which is then used for a real-time warning system and for a hazard assessment ( i.e. a long term hazard assessment). A better title would be 'Development of a Tsunami Warning System and Hazard Assessment for the coast of Oman'.

2. Line 36: It would be useful to indicate the locations of Muscat and the Gulf of Oman on Figure 1, for readers who are not familiar with the region.

3. Line 53: You note that tsunami warning systems are commonly built on pre-computed scenarios, but there are no references to back this up (although there are plenty of references to other hazard assessments in the next paragraphs). In order to better place this work in context, at least few other tsunami warning systems should be referred to and discussed. A starting point could be the various systems referred to within Greenslade, D. J. M., et al. (2014), An assessment of the diversity in scenario-based tsunami forecasts for the Indian Ocean, Cont. Shelf Res., 79, 36–45.

4. Line 145. You should refer to the other systems that this scenario database is similar to, rather than those it is different from (see references in paper above).

5. Section 2.2 How did you choose these 6 events? Did you make any effort to ensure that they are the worst cases, by, for example, examining the wave heights at the coast, or the run-up or inundation extents? Or is it just the highest magnitude earthquakes?

6. Line 304: How do you know that the return periods are ' thousands of years'?

7. Lines 370-371. It seems to me that a better description of the objective of this research is: (1) the development of a scenario database, in order to provide the basis for (2) a tsunami warning system and (3) a tsunami hazard assessment.

8. Line 392. How many of these validation points are there? Where are they? For how long have they been installed? Please indicate on one of the figures the locations of these tide gauges. A table with various details of the tide gauges might also be useful (i.e. lat, lon, name, depth, first installed etc.)

9. Line 391-395. Are there any examples of tsunamis that have been observed at any of these tide gauges? A model/obs comparison would be very useful for validation of the system.

10. Line 416/table 4. How were the amplitude thresholds for the different threat categories selected? What is the difference between T1 and T2 in the header of the table?

11. Even if it is not possible to do any validation of the tsunami amplitudes with tide gauge observations, it should be possible to validate the threat categories. In particular, it would be useful to see what the Threat categories would be for the 1945 event, and whether that matches with the known impacts.

12. Line 427. Figure 5 shows that there are some small areas of overlap in the coastal grids. Were the results identical within these overlap areas? If not, how did you select which grid to use?

Technical Comments

Line 9: developing = the development of

Line 10: as it is the case of = such as

Line 10: remove 'followed'

Line 11: remove ' obtained'

Line 16: propagated = modelled; feed = create; work = act

Line 17: case of tsunami = case of a potentially tsunamigenic earthquake

Line 29 occurred = occurring

Line 30: elaboration = development (and make this change at all other places in the text)

Line 30: on = for

Line 34: (NCSI, 2014) not in references

Line 39: recent = notable (1945 is not really recent)

Line 43: defend = suggest

Line 47: feeds the = supports a

Line 48: the tsunami hazard = a tsunami hazard

Line 50: the tsunami warning system = a tsunami warning system

Line 53: contain = are based on

Line 54: a new earthquake = a potentially tsunamigenic earthquake

Line 54: presents the closer = selects the closest

Line 61: the tsunami hazard = a tsunami hazard

Line 63: 'hazard degree for people instability'. It is not clear what this means (this phrase occurs in a few other places as well)

Line 97: scenarios database = scenario database (and elsewhere in the text)

Line 127: Strasser et al, 2010. The two Strasser papers in the reference list are identi-

cal.

Line 129: The Leonard et al 2014 paper should probably be Leonard et al., 2010. Leonard et al 2014 does not seem to be relevant to this work.

Line143: contemplate = cover

Line 157: Or = Either

Line 160: exclude = excluding

Line 262: Being the tectonic characteristics similar = Since the tectonic characteristics are similar

Line 358: Figure 5 and caption. It would be better to call the level 1 grid 'Regional' rather than 'Global', since it does not cover the entire globe.

Line 366. Jonkman et al reference should first appear here

Line 504-505: '. . .receive and then spread all the relevant data'. It is not clear what this means.

Line 519: 'aisled'? Not sure what this means in this context

Line 531: affection = impact (and elsewhere)

Line 624: You just need one Okada (1985) reference
* * *

---

## Referee Comment (RC2) · J. Behrens (Referee) · 22 Aug 2018

**1   General Comments**

The article describes the development of a tsunami scenario database for utilization in tsunami early warning and hazard assessment for the coast of Oman. From a seismic assessment of possible relevant scenarios and their determination in terms of parameters for computational fault planes, via numerical tsunami propagation and inundation modeling to the derivation of relevant hazard parameters and their visualization the authors cover a complete simulation workflow. It appears that their product is already

implemented in the operational tsunami warning system of the Sultanate of Oman.

While this work is important and relevant for practical applications, I am afraid to say that scientifically I feel the article is not publishable at the current state. The reason for this evaluation is manyfold.

1. First and foremost, the description of the usage of scenarios in an early warning situation is not state of the art. It has been shown in previous work [Behrens et al., NHESS, 2010] - and I am a little embarrassed to mention my own work here - that the sensitivity regarding source parameters in the near field is so high that early assessment uncertainty in the seismic parameters can render a simple selection based on seismic location and magnitude useless at best and misleading in the worst case. The authors should know about these works and should consider corresponding consequences.

2. Second, the usage of a worst-case based hazard assessment is not state of the art. And in particular by selecting 7 of the more than 3000 scenarios as the potentially worst appears doubtful, since local amplification can lead to unsuspected effects of even smaller sized scenarios. So, even in a worst case assessment strategy, all scenarios should be aggregated and not just a few hand selected ones. Or it should be made reasonable by scientific evidence that this selection is sound.
In addition to this, it should be explained, why a SBTHA approach is chosen over a PTHA approach. What is the advantage in the particular setting?

3. Third, the use of a draft parameter using shallow water simulated onshore velocity data appears very doubtful. I am not aware of a scientific study that evaluates the reliability of velocity data of shallow water codes in the special situation of a wetting-and-drying region and personally I have doubts that these data are in any way realistic. So, their use in hazard maps should be scientifically assessed and motivated.

4. For the inundation simulations, the authors are not able to show validated reference scenarios. Do the inundation maps coincide with historic data (if available)? At least some data from the 1945 event in the Makran Subduction Zone should be available for validating the inundation length and flow depth. In fact, the common standards of validating operational tsunami simulation codes demand not only for a general test, but for validation with field data in the application area (see e.g. Synolakis et al., PAGEOPH, 2008). The reason for this validation step is that certain parameters need to be calibrated for the situation.

5. Furthermore, it appears that the composition of seismic scenarios by means of Okada plates in those complex non-subducting strike-slip fault regions away from the Makran Fault Zone is somewhat too simplistic. A more thorough assessment of the possible tsunami sources should be made.

6. Additionally, much more care should be taken to writing and text structuring. The nomenclature is often inconsistent (e.g. *Mw* is used as well as $M_w$), sentences are incomplete or redundant, and a native speaker should revise the text for proper language usage and correct the large number of typos.

**2 Specific Comments**

P.1 L.26 The claim that tsunamis pose a bigger threat than earthquakes, hurricanes and tornadoes needs scientific justification by references or numbers. I doubt this. And if it is true, you need to define the metric in which you compare (loss of life, property, ...)

P.2 L.41 Please use unanimous nomenclature for Mw (see abstract).

P.2 L.46 This sentence is not clear to me.

P.4 L.111 ff. It is not clear to me, why recent seismicity should reveal the structure of future worst cases. You should provide scientific references that prove this claim.

P.4 L.118 This last sentence of the paragraph is a repetition of a previous one.

P.6 L.145 ff. When differentiating your scenario approach from a unit source approach, then it would certaintly be appropriate to mention also other approaches. First of all, it would be appropriate to mention [Babeyko et al., NHESS, 2010] who use a similar but more sophisticated constrained scenario approach in the GITEWS/InaTEWS system. Secondly, the JMA approach should also be mentioned where every Okada parameter is varied at each location, yielding 100thousands of scenarios.

P.6 L.172 You should conduct sensitivity studies, since it appears that in particular representing the large Mw8.5 and larger scenarios by just one Okada plate in the near field is not appropriate and yields unrealistic wave patterns. How does this effect local inundation and wave parameters is a necessary exercise in order to assess the uncertainty from the modeling approach.

P.8 L.200 f. Could you describe how you represent the more oblique strike-slip and reverse fault events by means of Okada plates? Is this realistic?

P.10 L.227 ff. What does $N70 - N90°E$ mean, is it North or East?

P.10 L.250 The inversion to obtain magnitude from the size of a plate via scaling laws is interesting and I did not know about this. But is there some reference that introduced this method? If not, then it should be explained in more detail and in particular it should be investigated more solidly by sensitivity analysis.

P.11 L.256 ff. The language in that paragraph needs major revision.

[Figure]

P.14 L.331 ff. Please describe in detail: Is the sea floor bathymetry AND the sea surface changed by the initial condition? And do you account for dispersion of the sea floor displacement in the water column?

P.14 L.337 ff. Please describe in more detail: How did you merge the bathymetry and topography maps, since most of the times sea level zero is different in these communities. Did you find any problems with this? Furthermore, what was the rationale behind chosing the named resolutions? Is 45 m on the finest level enough to compute realistic inundation maps? Did you perform sensitivity analysis?

P.16 L.367, Fig. 6 You give different definitions for the drag level: either $d = u \cdot h$ or $d = u \times h$. Since $u$ is a vector and $h$ is a scalar, these definitions are somewhat ambiguous, anyways. What exactly do you mean?

P.16 L.370 f. This sentence is incomplete.

P.16 L.373 This sentence is redundant, since the exact same informaton was given before. It would be more interesting at this point to get more information about the modeling setup, e.g. time step size, bottom roughness parameterization, dispersive/non-dispersive equation set, linear/non-linear equations, etc.

P.16 L.379 What are graphical epicenter locations versus numerical ones?

P.16 L.380 It was already mentioned in the general comments above that the selection mechanism is much too simple for the near field. This is scientifically not sound.

P.17 L.391 ff. What you call validation point is somewhat missleading. Validation is the process of checking if a simulated result represents the physical reality. This is not intended in an early warning situation. You want to *compare* or *match* the values from measurements to simulation results. So, I would suggest to rename these points.

[Figure]

P.20 L.422 ff. These two sentences contain no new information and can be cancelled without loss.

P.20 L.434 While reading the article in one piece, at this point I was not aware any longer that you had selected 7 scenarios for your worst case aggregation. So, the number 7 here falls a little bit out of heaven. Maybe you could add a paragraph or two, explaining in more detail your procedure to select those 7 scenarios at this point.

P.22 L.445 f. It is not clear to me, what you mean by "... using the equinoctial highest tidal level as reference". Do you mean you set this water level to be the still water level for your simulations? Or do you compare the inundation height/runup-level obtained with a mean sea level zero with this equinoctial tide level? Please explain.

P.22 L.469 Can you please define the term *people instability*?

P.28 L.576 The reference Fernandez et al. is formatted incorrectly 'amp;' and is this reference peer reviewed?

P.30 L.645 ff. The reference to Strasser et al. appears twice.

---

## Referee Comment (RC3) · C. Moore (Referee) · 31 Aug 2018

General Comments:

An interesting description of a warning system for Oman, as well as a hazard assessment using a deterministic approach. The methodology for developing the scenario database based on the investigation of regional seismicity, and the model used for creating the database both seem fairly simplistic, though perhaps adequate. The language used in the paper is fairly poor, and, while I don't think it detracts much from the submission, I do think a revision should include more attention to grammar. More of a concern might be the overall accuracy of the designed warning system, and the

limited number of sources used in the hazard assessment. I'll leave the decision as to whether there is enough new material here to warrant publication to the reviewers.

Specific Comments:

While a scenario database catalog has proved a useful tool for NOAA's tsunami warning system, an important difference between the NOAA system and the one described in this paper is in the direct deep-water measurement of the tsunami. This measurement data during a tsunami constrains the scenario output, compensating somewhat for inevitable inaccuracies in seismic parameters, particularly in the far-field.

The authors mention choosing the "closer pre-computed scenario" (line 54), but neglect to elaborate on how: the closest epicenter, or are different epicenters used for different magnitudes? How will the epicenter and magnitude be obtained, and what is the error in each of these likely to be? Small errors in these parameters (not to mention variations in strike, dip, and rake) can lead to large errors in inundation and wave height estimates in the near field. While refinements in epicenter and magnitude estimates during an event can result in smaller errors, these can take time: perhaps a discussion of estimates of the time it takes to obtain these from seismometer data, and what effect errors in epicenter would have on the forecast might help for this paper, and for the forecast system designed.

A mention of the 1945 event was made, but no mention of the data collected during that event (there is both tide gauge and inundation witness data available), or any attempt at validating the model with data. Even a comparison with anecdotal data can help when it comes to validation, and without it the modeling accuracy is unknown.

Lastly, if 3181 scenarios were run for the database, why were only 7 chosen for the assessment? Since the assessment was a simple deterministic approach, and a composite was made, perhaps all (or at least a larger number) might be used for the composite.

---

## Referee Comment (RC4) · Anonymous Referee #4 · 2 Sep 2018

**Review of the paper entitled "Tsunami Hazard assessment and Scenarios Database for the Tsunami Warning System for the coast of Oman", by Aniel-Quiroga et al. 2018**

The paper is a Tsunami Hazard assessment for the coast of Oman. It generates large Tsunami database Scenarios for the sake of Tsunami Warning System (TWS). The paper treats all the seismic sources as if they are tsunami generating sources even if there is no possibility or evidence of the sources to generate Tsunami, and this may be client oriented for completeness of TWS data. It uses sound methodology for tsunami simulation followed by discussion and conclusions. References are adequate. In its current form, it make an excellent report to client as it legally cover the authors from the fear if something happens. It hide behind fear if something happen rather than give scientific evidence of the choices of large magnitudes. It can be reduced to be a good scientific paper when it treats the tsunamigenic seismic sources scientifically with supporting evidence of the magnitudes.

**Detailed comments:**

1. Line 36: Oman is mostly affected by the Makran trench: does Makran has a trench? Also is it Gulf of Oman to Sea of Oman -consistent?
2. Lines 38 and 39: "significantly impacted on several countries" need revision
3. Line 40: Run -ups between 5 and 10 meters: is this observed on the current coast of Oman (not used to be)? (Give reference).
4. Line 44: which then arrived delayed to the coast of Oman change to which explains the delayed arrival to the coast of Oman
5. Lines 116-119: need more detail on the database for each seismic area
6. Line 146: change were define to were defined
7. Line 157 -158: need revision to be understood?
8. Lines 166 and 167: width (24) > length (20)?
9. Lines 180-184: Is Blaser et al, 2010 appropriate to use here? What is the maximum fault length below water in this particular area? How could we determine the rupture here while this zone is characterized by blind faults? Historically, No earthquake with magnitude greater than 7.0 occurred in this particular area?
10. Lines 192-198: width of straight has nothing to do with the fault length.
11. Lines 220-223 and 232: is Blaser et al. 2010 subduction relation appropriate to use here? Compare the rupture length and maximum magnitude with that of Zagros.
12. Lines 247-249: does Strasser et al., 2010a (interface and intraslap subduction) relation appropriate to use here?
13. Line 282: latitude and depth how the depth is computed? Dip in north Makran is different from south Makran? Do the author need to use additional scenarios at different depths along the subduction zone?
14. Line 298: if both segments break in one earthquake: WHY?
15. Lines 205-306: the 2004 tsunami in Indonesia and Tohoku, Japan in 2011? Compare the comparable? In what sense do authors compare Makran subduction to these

subductions i.e. seismicity rate?, convergence rate?, trench shape?, age of subduction slap? Etc.

16. Line 445-446: values of the hazard and specific area in the northern coast?

17. Discussion: it would be nice to compare the national vs local hazard maps of some common parameters and some location points so to show the effect of DEM input resolutions.

18. Figures: most are unclear and blurry (writing, scale, lat. Lon. unreadable)

    a. Fig. 2: all epicenters of the scenarios are at the center of the rectangles, the 1945 earthquake shows a unilateral rupture toward the east (Byrne et al. 1992). Should the authors postulate unilateral rupture toward east for some additional scenarios?

    b. Fig. 3: Why no more than one segment can break in one earthquake as the case of Makran?

    c. Fig. 7: not readable and unclear

    d. Fig. 8: for which scenario is this graph (magnitude and distance)

    e. Fig. 10: is an important figure and need to be in higher resolution

    f. Fig. 11: is an important figure need to be in high resolution and increase the sizes similar to 11a

    g. Fig 12: similar to figure 11, increase the sizes of the subsequent figures like fig 12a

**References**

Byrne D E, Sykes L R & Davis D M (1992) Great thrust earthquakes and aseismic slip along the plate boundary of the Makran subduction zone. J Geophys Res 97:449-478.

Deif A, Al-Shijbi Y, El-Hussain I, Ezzelarab M, Mohamed AME, 2017. Compiling an earthquake catalogue for the Arabian Plate, Western Asia, Journal of Asian Earth Sciences 147 (2017) 345–357. http://dx.doi.org/10.1016/j.jseaes.2017.07.033

El-Hussain I, Deif A, Al-Shijbi Y, Ezzelarab M, Mohamed AME, 2018. Developing a seismic source model for seismic hazard studies in the Arabian Plate, Arabian Journal of Geosciences (11):435, http://doi.org/10.1007/s12517-018-3797-7.